# Alternative Substrates and Fertilization Doses in the Production of *Pinus cembroides* Zucc. in Nursery

**Rosa Elvira Madrid-Aispuro** [1], **José Ángel Prieto-Ruíz** [2,*], **Arnulfo Aldrete** [3],
**José Ciro Hernández-Díaz** [4], **Christian Wehenkel** [4], **Jorge Armando Chávez-Simental** [4] and
**John G. Mexal** [5]

1. Programa Institucional de Doctorado en Ciencias Agropecuarias y Forestales, Universidad Juárez del Estado de Durango, Constitución 404 sur Zona centro, 34000 Durango, Mexico; madrid.rosa@ujed.mx
2. Facultad de Ciencias Forestales, Universidad Juárez del Estado de Durango, Río Papaloapan y Boulevard Durango S/N Col. Valle del Sur, 34120 Durango, Mexico
3. Colegio de Postgraduados, Campus Montecillo, Carretera México-Texcoco Km 36.5, Montecillo, Texcoco, 56230 Estado de México, Mexico; aaldrete@colpos.mx
4. Instituto de Silvicultura e Industria de la Madera, Universidad Juárez del Estado de Durango, Avenida Veterinaria y Boulevard del Guadiana 501, Circuito Universitario, Torre de Investigación, 34120 Durango, Mexico; jciroh@ujed.mx (J.C.H.-D.); wehenkel@ujed.mx (C.W.); jorge.chavez@ujed.mx (J.A.C.-S.)
5. Plant & Environmental Sciences. New Mexico State University. P.O. Box 30003, Las Cruces, NM 88003-8003, USA; jmexal@gmail.com
* Correspondence: jprieto@ujed.mx; Tel.: +52-618-827-12-20

**Abstract:** Rooting substrate and fertilization are key components in the production of containerized seedlings, as they can influence the morphological and physiological characteristics of the plants, which in turn can impact outplanting performance. The objective of the study was to evaluate the effect of four substrates based on mixtures of peat moss (PM), composted bark (CB) and raw pine sawdust (PS), combined with two doses of controlled release fertilizer (CRF) and one non-fertilized control, on the growth of *Pinus cembroides* Zucc. in the nursery. The treatments were: M1: 50+25+25, M2: 25+25+50, M3: 25+50+25 and M4: 50+50+0 of PM+ CB+ PS (% by volume), respectively. Fertilizer treatments used a controlled release fertilizer (Multicote®): F1: 3 kg m$^{-3}$ and F2: 6 kg m$^{-3}$ and a control (WF: with no added fertilizer). The treatments were distributed in a randomized complete block design, with a factorial arrangement of 4 × 3 and six replications. The variables evaluated were: height, seedling diameter, dry biomass, Dickson Quality Index, N, P and K content. Regardless of the substrate, the high fertilizer dose (6 kg m$^{-3}$) improved most morphological variables. In addition, the high fertilizer dose resulted in foliar N, P and K concentrations within recommended ranges for all substrates. The substrate containing only peat moss and composted bark (M4 + F1 and M4 + F2) had the best growth response. However, the substrate composed of 25% peat moss, 50% composted bark and 25% raw pine sawdust with the high fertilizer dose (M3 + F2) resulted in acceptable seedling growth, and may be preferred if the cost of the substrate is a concern to nursery manager.

**Keywords:** controlled release fertilizer; raw pine sawdust; composted bark; peat moss; plant quality; nutrition

## 1. Introduction

The loss of natural resources and the degradation of ecosystems is a worldwide problem, but it is particularly problematic in Mexico, which is one of the most important centers of biological diversity [1]. In recent years the disturbance of forest ecosystems through both deforestation and forest degradation has been significant, and has endangered both the resilience and sustainability of these forests [2].

In response, Mexico has implemented reforestation programs for conservation, restoration and commercial plantations. In the last five years, 144 million seedlings have been produced annually to reforest 112,000 ha yr$^{-1}$ [3]. However, the average survival rate after one year of planting has been less than 50% due primarily to drought, planting at the end of the rainy season and poor-quality seedlings [4,5]. Therefore, at least one part of the solution is to produce nursery seedlings with the morphological and physiological attributes necessary to resist the initial stress after planting and ensure its subsequent growth in the planting site [6,7]. Equally important is to produce these seedlings in a timely fashion to take advantage of the rainy season which could ameliorate the effects of subsequent drought.

Evaluating the morphological and physiological characteristics of nursery plants provides an estimate of the survival and growth potential of those plants following outplanting [8,9]. Diameter is the morphological variable that most correlates with field survival [10,11], but other important variables are total dry weight and the Dickson quality index [12]. Furthermore, these indicators will certainly vary among species, as well as site characteristics, viz. dry versus mesic sites [13].

Cultural practices in nursery influence the morphological characteristics of seedlings [7], including irrigation, fertilization, substrate composition, container size and type. Substrates greatly affect the growth and development of seedlings [14]. Peat moss, agrolite and vermiculite have been used for a long time in numerous countries; however, availability is limited, and they can be expensive, especially if imported. In addition, harvesting peat causes ecological impacts [15–17]. In recent years, alternative materials have been evaluated as a substrate. These are mainly organic by-products from non-forest industries [18–22]. An alternative to reduce the use of imported substrates is to use local and regionally available materials, which can be less expensive, sustainable, and suitable for high seedling quality production [23].

Other materials that can be used as a substrate are by-products of the forestry industry. They are often abundant and cheaper than peat moss [24]. Various combinations of materials have been used, since as yet there is no single ideal substrate for all species. Nevertheless, an adequate substrate must fulfill basic physical characteristics, including porosity, adequate water retention capacity and providing the plants with physical support [25,26]. In addition, important chemical characteristics including pH, electrical conductivity, cation exchange capacity, organic matter and carbon/nitrogen ratio must be considered [26].

In the last few decades, pine bark has been used successfully in Mexico and U.S.A. Due to its physical properties of high porosity and adequate drainage, it is now considered an important component for the production of containerized seedlings [21,27,28]. Similarly, raw pine sawdust has been used successfully as a component in combination with pine bark and peat moss, to produce *Pinus* seedlings [22]. Sawdust improves moisture retention in a substrate, is inexpensive and widely available. Results of recent studies confirm that sawdust, in combination with other materials is a suitable substrate in *Pinus montezumae* Lamb., *Pinus pseudostrobus* Lindl., *Pinus cooperi* Blanco and *Pinus engelmannii* Carr. [22,29–31].

Mineral nutrition is another important factor in nurseries production [32] and can be supplied to seedlings by incorporating the fertilizer into the substrate during its preparation (usually controlled release fertilizer) or through the irrigation (fertigation). In addition, fertilization dosage and frequency should be considered. The use of controlled release fertilizers in the substrate increases the efficiency in the supply of nutrients to the plants, with prolonged release times, which reduces the loss of nutrients by leaching or overspray [33,34].

*Pinus cembroides* Zucc. it is one of the most important conifer species in Mexico; it is distributed naturally from the state of Puebla, Mexico, to Arizona in the southwestern United States [35,36]. It inhabits sites with altitudes of 1350 to 2800 m, with annual rainfall of 350 to 700 mm. It is widely used in reforestation programs in semi-arid regions [37]. However, to the best of our knowledge, there have been little research on *P. cembroides* seedling production.

The objective of this study was to evaluate the effect of four substrates consisting of peat moss, composted bark and raw pine sawdust, combined with two doses of controlled release fertilizer and an unfertilized control, on the morphological growth and assimilation of nitrogen (N), phosphorus (P), and potassium (K) in *Pinus cembroides* Zucc. under nursery conditions. We started with the hypothesis that the use of raw pine sawdust and composted bark, mixed with peat moss, in combination with controlled release fertilizer and water-soluble fertilizer, would be suitable for the optimal growth of *P. cembroides* seedlings in the nursery.

## 2. Materials and Methods

### 2.1. Study Site

The experiment was conducted in the "General Francisco Villa" forest nursery, located in Ejido "15 de Septiembre" in the municipality of Durango, state of Durango, Mexico, located at coordinates 23°58′21″ N, 104°35′56″ W at an altitude of 1875 m. The seedlings were produced in a greenhouse covered with 720-gauge polyethylene plastic, treated against ultraviolet rays and with a 30% shade mesh. During the development of the plants, a maximum average temperature of 40 °C and a minimum average temperature of 3 °C were recorded, while the average relative humidity was 70%. The average light intensity was 370 klx (Figure 1).

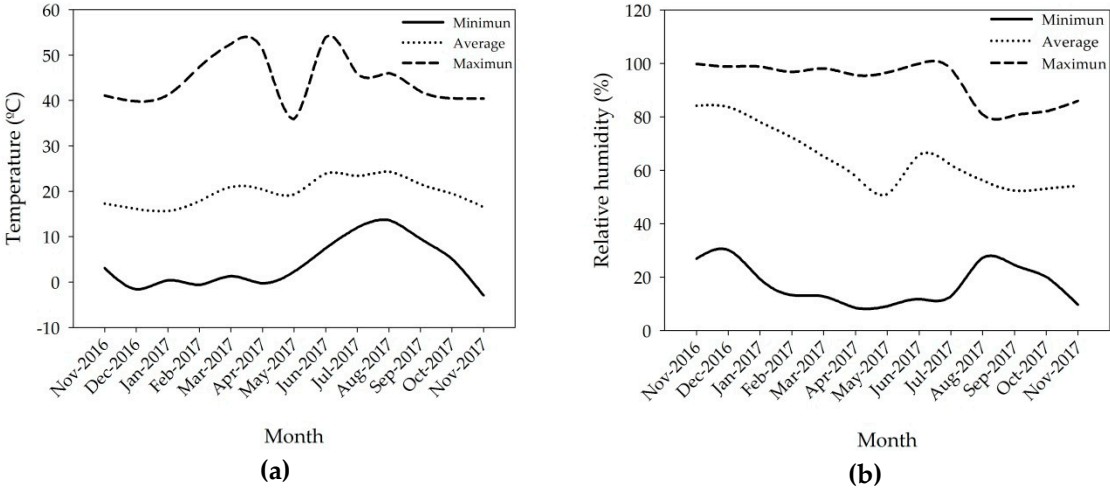

**Figure 1.** Environmental conditions recorded during the production of *Pinus cembroides* Zucc. from November 2016 to November 2017: Temperature (**a**) and relative humidity (**b**).

### 2.2. Establishment of the Experiment

The seedling production cycle was from August 2016 to October 2017 (14 months). The sowing was undertaken in seedbeds, in a mixture in equal parts of composted bark and peat moss. Four weeks later, 5544 plants were transplanted into polystyrene trays of 77 cavities with a capacity of 170 mL per cavity, with an upper diameter of 4.3 cm and a length of 15 cm. Before sowing, seeds were soaked in water for 24 h at a temperature of 20 to 25 °C and disinfected for 10 min with a 10% commercial bleach solution [38]. The seeds were air-dried and treated with the fungicide (Tecto 60® Tiabendazol: 2-(4-thiazolyl)-1H-benzimidazole, Syngenta, Mexico City, Mexico).

## 2.3. Treatments

The combination of four substrates and two fertilization doses and one unfertilized control were evaluated. The substrates were mixtures of peat moss (PM), composted bark (CB) and pine raw sawdust (PS), in the following proportions by volume: M1: 50+25+25, M2: 25+25+50, M3: 25+50+25 and M4: 50+50+0, of PM+CB+PS, respectively. The fertilizers and doses applied were: WF = without controlled release fertilizer (control), F1 = 3 kg m$^{-3}$ and F2 = 6 kg m$^{-3}$ of the controlled release fertilizer Multicote 8® 18% N + 6% P$_2$O$_5$ + 12% K$_2$O (Haifa Chemicals Ltd., Haifa, Israel) with eight to nine months nutrient release (Table 1). The M4 substrate is the basic mixture currently used in Mexican nurseries. In the mixtures M1, M2 and M3, sawdust was proposed as a cheaper alternative substrate to decrease the proportion of the peat moss in the mixture.

**Table 1.** Substrates and fertilization doses evaluated in the production of *Pinus cembroides* Zucc. under nursery conditions.

| Substrate | Component (%) | | | Fertilization | | Treatment |
|---|---|---|---|---|---|---|
| | Peat Moss | Composted Bark | Pine Raw Sawdust | | Multicote® (kg m$^{-3}$) | |
| M1 | 50 | 25 | 25 | WF | - | M1 + WF |
| M1 | 50 | 25 | 25 | F1 | 3 | M1 + F1 |
| M1 | 50 | 25 | 25 | F2 | 6 | M1 + F2 |
| M2 | 25 | 25 | 50 | WF | - | M2 + WF |
| M2 | 25 | 25 | 50 | F1 | 3 | M2 + F1 |
| M2 | 25 | 25 | 50 | F2 | 6 | M2 + F2 |
| M3 | 25 | 50 | 25 | WF | - | M3 + WF |
| M3 | 25 | 50 | 25 | F1 | 3 | M3 + F1 |
| M3 | 25 | 50 | 25 | F2 | 6 | M3 + F2 |
| M4 | 50 | 50 | - | WF | - | M4 + WF |
| M4 | 50 | 50 | - | F1 | 3 | M4 + F1 |
| M4 | 50 | 50 | - | F2 | 6 | M4 + F2 |

M1, M2, M3 and M4 are the substrates mixtures of peat moss + composted bark + pine raw sawdust in the indicated proportions. WF = without controlled release fertilizer (control), F1 = 3 kg m$^{-3}$ and F2 = 6 kg m$^{-3}$ of the controlled release fertilizer Multicote 8® 18% N + 6% P$_2$O$_5$ + 12% K$_2$O.

In addition to the substrate fertilizer treatments, water-soluble fertilizers were applied in the following doses: from November 2016 to February 2017: Poly feed® 8% N + 52% P$_2$O$_5$ + 17% K$_2$O + 1000 ppm Fe + 500 ppm Mn + 200 ppm B + 150 ppm Zn + 70 ppm Mo (Haifa Pioneering the Future, Haifa, Israel) and Ultrasol® SOP 51% K$_2$O + 18% S (SQM The Worldwide Business Formula, Taiwan) were applied at 0.5 g L$^{-1}$ each. Between March and June 2017, Peñoles® Ultrasoluble Ammonium Sulphate 20% NH$_4$ + 23% S + 0.60% K$_2$O + 190 ppm minor elements (Fertirey Peñoles, Torreon, Coahuila, Mexico) were applied at 1.5 g L$^{-1}$. From July to October 2017, all the plants were fertilized with Ultrasol® SOP 51% K$_2$O + 18% S at 0.5 to 1.5 g L$^{-1}$. In each irrigation, applied every third day, the plants received 35 mL of water with fertilizer.

## 2.4. Experimental Design

The experiment was established in a randomized complete block design, with a factorial arrangement (four substrates x two doses of controlled release fertilizer and one control), with six blocks per treatment. The experimental unit was a tray with 77 plants, out of which the 45 central plants were evaluated to avoid possible edge effects.

## 2.5. Measured Variables

After 14 months of plant growth, 12 seedlings were randomly extracted per experimental unit (72 plants per treatment). Shoot height (cm) and ground line diameter (mm) were measured. Roots were excavated from the growing medium by washing with water. Root volume was determined by the water displacement method [39]. Shoots were separated from roots, placed in paper bags and then

in a FELISA® FE-291D drying oven for 72 h at 70 °C and weighed on an Ohaus® PA214 analytical balance (Ohaus of Mexico City, Mexico). The Dickson Quality Index (DQI) [12] was calculated:

$$DQI = \frac{Total\ dry\ weight\ (g)}{\frac{Height\ (cm)}{Diameter\ (mm)} + \frac{Stem\ shoot\ part\ dry\ weight\ (g)}{Root\ dry\ weight\ (g)}} \tag{1}$$

To estimate nutrient concentrations (N, P and K), needles (5 g) were collected from seedlings in the center of all trays of each treatment (three repetitions per treatment). Nitrogen was determined through the Kjeldahl method [40], by photocolorimetry, and by reduction with Vanadato-Molybdate. The concentrations of phosphorus and potassium were estimated through the direct reading of the digestate in an atomic absorption spectrometer [41]. Analyses were carried out in the laboratory of the National Center for Disciplinary Research on the Relation of Water, Soil, Plant, Atmosphere (CENID-RASPA) of the Mexican National Institute of Agricultural and Livestock Forestry Research (INIFAP). Physical characteristics of the substrates were determined, including: aeration porosity (%), water holding porosity (%) and total porosity (%), using the method described by Landis [42].

*2.6. Statistical Analyses*

Due to the non-compliance of the normality assumptions with the Shapiro–Wilk and Kolmogorov–Smirnov tests, the data of the morphological variables were evaluated by using the non-parametric statistical test of Kruskal-Wallis [43] with the PROC NPAR1WAY procedure; a Bonferroni-Dunn means separation test ($\alpha = 0.05$) was used [44]. The values of N, P and K underwent analysis of variance (ANOVA), as well as means tests, using the Tukey test. All statistical analyses were conducted using the software SAS version 9.2 (SAS Institute, Cary, NC, USA) [45].

## 3. Results

*3.1. Physical Characteristics of Substrate Mixtures*

The aeration porosity varied from 28.9% in M1 to 34.5% in M3. Moisture retention capacity ranged from 31.4 in M4 to 39.5% in M2, while the total porosity ranged from 65.0% in M1 to 72.4% in M2. Based on the reference values suggested by Landis [42], all the substrates met the suitability criteria of porosity, which means that all of them can technically be used as substrates (Table 2).

**Table 2.** Physical properties (mean ± standard error) of the substrate mixtures used in the production of *Pinus cembroides* Zucc. in nursery.

| Substrate | Aeration Porosity (%) | Moisture Retention Capacity (%) | Total Porosity (%) |
|---|---|---|---|
| M1 | 28.9 ± 0.7 | 36.1 ± 0.9 | 65.0 ± 1.0 |
| M2 | 32.9 ± 0.8 | 39.5 ± 1.7 | 72.4 ± 1.0 |
| M3 | 34.5 ± 0.6 | 35.6 ± 2.2 | 70.1 ± 1.9 |
| M4 | 33.9 ± 1.3 | 31.4 ± 1.6 | 65.3 ± 0.6 |
| * Reference value | 15–35 | 25–55 | 60–80 |

M1 = Peat moss (50%) + composted bark (25%) + raw pine sawdust (25); M2 = Peat moss (25%) + composted bark (25%) + raw pine sawdust (50%); M3 = Peat moss (25%) + composted bark (50%) + raw pine sawdust (25%); M4 = Peat moss (50%) + composted bark (50%). * Reference value [42].

*3.2. Morphological Characteristics and Quality Index*

Substrate, fertilizer, and interaction significantly ($p = 0.00416$) affected the morphological variables evaluated. In general, the peat moss and bark (1:1) substrate produced larger plants. Furthermore, controlled release fertilizer significantly improved growth compared to the control (WF), regardless of the substrate (Table 3).

**Table 3.** Average values (mean ± standard error) for the morphological variables by treatments, in the growth of *Pinus cembroides* Zucc. after 14 months under nursery conditions.

| Treatments | Height (cm) | Diameter (mm) | Dry Biomass (g) | | | Root Volume (cm$^{-3}$) | Dickson Quality Index |
| | | | Shoot | Root | Total | | |
|---|---|---|---|---|---|---|---|
| M1 + WF | 10.3 ± 0.2f | 3.20 ± 0.1c | 1.35 ± 0.1d | 0.41 ± 0.02d | 1.76 ± 0.1e | 1.2 ± 0.1d | 0.27 ± 0.01d |
| M1 + F1 | 15.9 ± 0.4d | 4.50 ± 0.1a | 2.31 ± 0.1bc | 0.79 ± 0.03abc | 3.10 ± 0.1bc | 2.2 ± 0.1bc | 0.48 ± 0.02ab |
| M1 + F2 | 17.9 ± 0.3ab | 4.51 ± 0.1a | 2.45 ± 0.1ab | 0.75 ± 0.03bc | 3.20 ± 0.1abc | 2.2 ± 0.1bc | 0.44 ± 0.02bc |
| M2 + WF | 9.5 ± 0.2f | 3.19 ± 0.1c | 1.31 ± 0.0d | 0.43 ± 0.02d | 1.73 ± 0.1e | 1.3 ± 0.1d | 0.29 ± 0.01d |
| M2 + F1 | 15.1 ± 0.3d | 4.32 ± 0.1ab | 2.10 ± 0.1c | 0.73 ± 0.02c | 2.82 ± 0.1c | 2.1 ± 0.1c | 0.45 ± 0.02bc |
| M2 + F2 | 18.1 ± 0.3ab | 4.31 ± 0.1ab | 2.51 ± 0.1ab | 0.81 ± 0.03abc | 3.32 ± 0.1ab | 2.4 ± 0.1ab | 0.46 ± 0.02ab |
| M3 + WF | 10.8 ± 0.3f | 3.48 ± 0.1c | 1.36 ± 0.1d | 0.46 ± 0.02d | 1.82 ± 0.1ed | 1.3 ± 0.1d | 0.30 ± 0.01d |
| M3 + F1 | 16.2 ± 0.3cd | 4.50 ± 0.1a | 2.43 ± 0.1abc | 0.81 ± 0.02abc | 3.24 ± 0.1abc | 2.3 ± 0.1bc | 0.49 ± 0.02ab |
| M3 + F2 | 17.5 ± 0.3bc | 4.53 ± 0.1a | 2.55 ± 0.1ab | 0.86 ± 0.05ab | 3.41 ± 0.1ab | 2.2 ± 0.1bc | 0.50 ± 0.02ab |
| M4 + WF | 12.4 ± 0.2e | 4.04 ± 0.1b | 1.66 ± 0.1d | 0.55 ± 0.02d | 2.21 ± 0.1d | 1.5 ± 0.1d | 0.37 ± 0.02cd |
| M4 + F1 | 18.0 ± 0.3ab | 4.69 ± 0.1a | 2.69 ± 0.1a | 0.90 ± 0.03a | 3.59 ± 0.1a | 2.7 ± 0.1a | 0.53 ± 0.02a |
| M4 + F2 | 19.2 ± 0.4a | 4.58 ± 0.1a | 2.74 ± 0.1a | 0.85 ± 0.03abc | 3.59 ± 0.1a | 2.3 ± 0.1bc | 0.48 ± 0.02ab |

Different letters indicate significant differences among treatments (Bonferroni-Dunn, $\alpha = 0.05$). M1 = Peat moss (50%) + composted bark (25%) + raw pine sawdust (25); M2 = Peat moss (25%) + composted bark (25%) + raw pine sawdust (50%); M3 = Peat moss (25%) + composted bark (50%) + raw pine sawdust (25%); M4 = Peat moss (50%) + composted bark (50%).

Treatments without the controlled release fertilizer incorporated into the substrate produced the shorter seedlings regardless of substrate mixture. The range across treatments was 1.3 cm, with the best response in substrate M4 at the two fertilizer doses. Substrate had no significant impact on seedling diameter, but substrate fertilizer had a significant effect on diameter growth. Dosage rate was not a significant factor in diameter (Table 3).

Biomass production was greatest with the use of peat moss and bark (M4) plus the controlled release fertilizer. There was little difference between fertilizer rates across substrates. Regardless of the substrate mix, the greatest shoot, root and total dry weights were obtained in the preplant fertilizer treatments (F1 and F2). While not always significant, it appeared that substrates with incorporated sawdust benefited by the high dose of fertilization (6 kg m$^{-3}$). There were no significant differences in the M4 treatment between fertilizer dosage. Root volume followed similar trends to other morphological parameters. The DQI variable provided little useful information other than the benefit of the incorporated fertilizer (Table 3).

### 3.3. Concentration of N, P and K in the Foliage

Nitrogen (N) concentration in the foliage varied from 2.20% to 2.84%, with no significant differences among treatments (Figure 2a). All treatments were well above the minimum N concentration recommended for conifers [46]. In contrast, there were significant differences in phosphorus (P) concentration, especially across fertilizer treatments. Phosphorus concentration increased from 0.36% in WF to 0.44% in F1 and 0.56% in F2 (Figure 2b). While all treatments were well above recommended P levels, the increased nutrition status could improve outplanting performance. The highest concentration of potassium (K) was obtained also in the high fertilizer treatments (F2), while the lowest values were recorded in treatments without controlled release fertilizer (Figure 2c). Values ranged from 0.66% to nearly 1.08%.

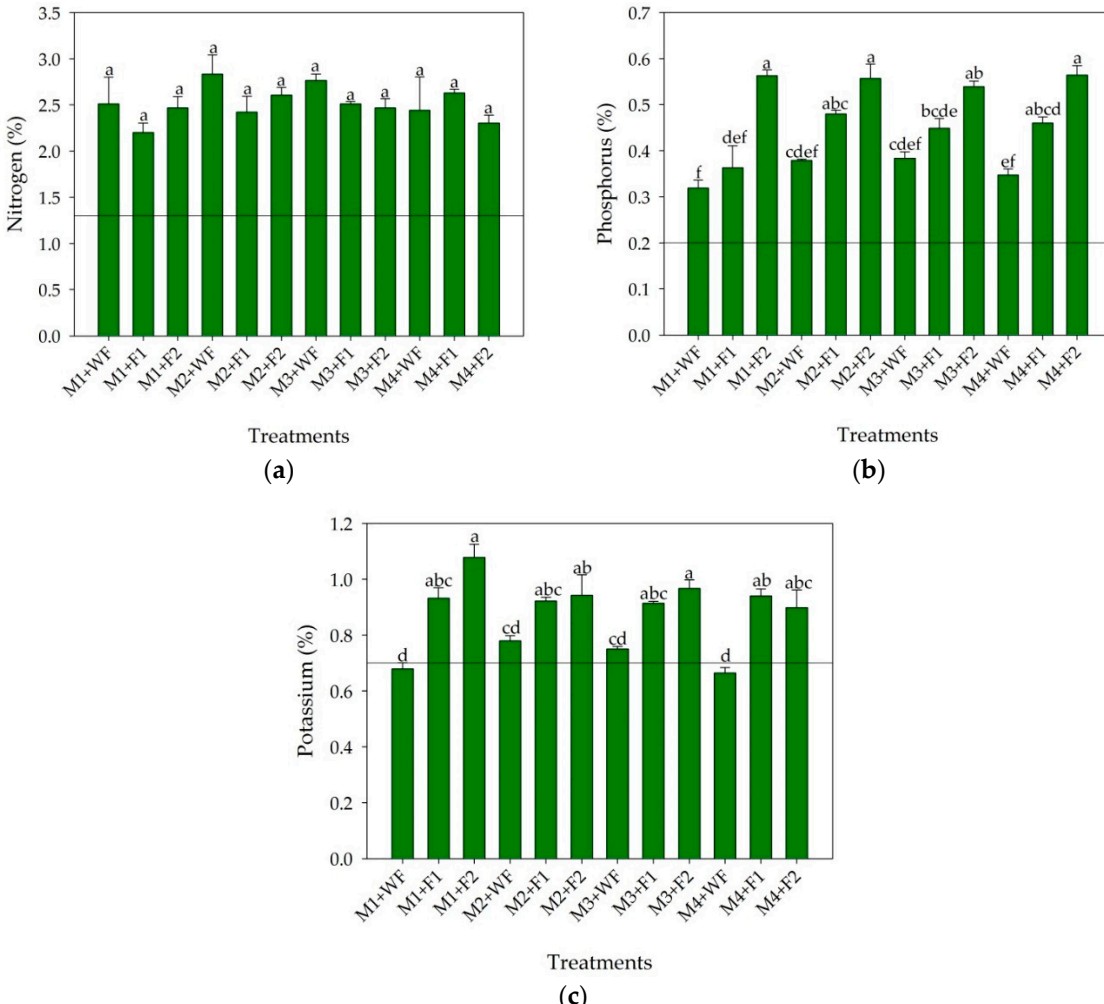

**Figure 2.** Fertilization settings in the foliage of *Pinus cembroides* Zucc. after 14 months under nursery conditions. Average concentration of: Nitrogen (**a**), Phosphorus (**b**) and Potassium (**c**). The horizontal lines indicate minimum values recommended for conifers [46].

## 4. Discussion

### 4.1. Morphological Characteristics and Quality Index

Based on criteria established in the Mexican Standard NMX-AA-170-SCFI-2016 [47], which indicates that the minimum height and diameter for *Pinus cembroides* should be 15 cm and 4 mm, respectively, all plants which received preplant controlled release fertilizer exceeded these parameters.

A quality plant has the ability to establish itself quickly and to initiate rapid growth in the field following outplanting [48]. Taller plants may have a competitive advantage in planting sites with heavy weed competition [49]. In this study, the tallest plants were with the highest fertilizer rate (Table 3). Similar results were reported by González et al. [22] in *Pinus cooperi* Blanco in substrate mixtures with similar proportions to those in this work. Also, Aguilera et al. [30] found that sawdust and composted bark can be used as a component of the substrate to produce *Pinus pseudostrobus* Lindl. seedlings. However, *Pinus cembroides* appeared to require greater fertilization with the addition of sawdust in the substrate, possibly due to the altered nutritional levels.

The plants remained in the nursery 14 months, which is the minimum time required to achieve target seedling size before being planted in the field. The controlled release fertilizer used in this experiment, has a nutrient release time of 8 to 9 months. Most controlled release fertilizers release a considerable percentage of nutrients shortly after application. Subsequent nutrient release is a function

of ambient temperature and substrate moisture. For this reason, Aguilera et al. [30] recommended a combination of controlled release fertilizers with different release periods for extended production cycles. This adjustment in fertilization could improve seedling growth over a long production cycle, especially if greater proportions of sawdust are incorporated into the substrate. The economies of using more sawdust have to be weighed against the costs of using higher rates and different release rates of fertilizers.

Diameter is an important predictor of survival and growth in the field, as it is a proxy indicator for the size of the root system [50]. Plants with a larger stem diameter tend to survive better than small stem diameter seedlings, which is why it is considered the most useful morphological measure of seedling quality [51]. Due to the above, in Mexico it is recommended that *Pinus cembroides* plants have a minimum diameter of 4.0 mm before being planted into the definitive site [47]. In this indicator, all treatments with fertilizer incorporated in the substrate achieved such minimum standard. On the other hand, the sawdust treatments with no slow release fertilizer failed to achieve the recommended minimum average diameter for this species (M1 + WF, M2 + WF, M3 + WF). Additionally, in conditions of zero fertilizer in the substrate the plant did not develop satisfactorily, since the substrates alone did not provide enough nutrients to the plant. It was noticed that when fertilizer was applied and the dose increased, the plants' diameter also increased, because the fertilizer provided not only additional nitrogen to the plant but also other nutrients necessary for its growth [52]. Niemiera et al. [53] noted that plants produced in wood-based substrates, require more fertilizer to equal the growth of plants grown in conventional organic substrates, such as bark and peat.

The stem, root and total dry biomass were larger in the plants receiving slow-released fertilizer, and biomass accretion appeared to be impacted by the addition of sawdust (Table 3). Treatments including sawdust in the substrate suffered a 20% reduction in biomass when no slow release fertilizer was incorporated, regardless of proportions. At the low rate of fertilizer (3 kg m$^{-3}$), total biomass was reduced by 16%, and only 8% at the high fertilizer rate (6 kg m$^{-3}$). This is further indication that additional fertilization is required when sawdust is incorporated into the substrate, as substrate porosity seemed unrelated to biomass response.

The DQI is a calculation incorporating the more common morphological attributes to estimate the probability of survival and growth after planting [54]. According to Tsakaldimi et al. [13] an increase of this index, represents better quality plants. Sáenz et al. [55] recommended DQI ≥ 0.50 for conifer species. In this case, the treatments with DQI values closest to 0.50 were, in general, the same, with better results in the other variables evaluated, characterized by having controlled release fertilizer and water soluble fertilizer.

Plants with higher root volume have higher survival and growth potential, since they are required for the absorption and transport of water and nutrients to the shoot [56,57]. *Pinus cembroides* characteristically has few lateral roots, but they are thicker which means they have larger reserves to grow, which can be useful in poor soils and with scarce precipitation (areas where this species commonly inhabits). Sawdust in WF treatments decreased root volume about 15%. However, at the high fertilizer rate, sawdust had little effect. Again, the M2 + F2 and M4 + F1 treatments had the highest root volumes, but there was little significant difference among treatments.

Raw sawdust has a high C:N ratio, which can negatively affect nutrient availability, especially nitrogen [19,58]. On the other hand, peat and composted bark not only have a relatively stable C:N ratio, but also have a high cation exchange capacity, which mitigates nitrogen immobilization and leaching of nutrients during irrigation, maintaining a high level of substrate fertility [59]. In the case of *Pinus engelmannii* Carr. up to 70% raw sawdust combined with composted bark and peat moss have been used favorably when 6 g L$^{-1}$ fertilizer dose of Multicote$^{®}$ was added [31]. These results are similar to ours.

*4.2. Nutrition*

The nutrient concentration N, P and K (Figure 1) were within the optimum reference ranges for conifers produced under nursery conditions [60]. The nitrogen concentration in the control treatment (WF) was statistically equal to treatments F1 and F2; this was due to the basic fertilization applied through irrigation. Landis [60] notes that foliar fertilization is used to correct deficiencies of micronutrients, such as iron deficiency chlorosis, but this can also be used to provide rapid "greening" before the plant is taken to the field. The reserves of stored nutrients are closely related to the survival probabilities and the growth of plants in the field [56,61]. Treatments M2 + F2 and M4 + F2 resulted in greater nitrogen and phosphorus concentration compared with those reported by González et al. [31] in *Pinus engelmannii* Carr. using the same substrates and fertilizer doses. Phosphorus content was especially responsive increasing from 0.35% in WF to 0.43% in F1 and 0.55% in F2.

Incorporating slow-release fertilizer in addition to water-soluble fertilizers produced greater root biomass and root volume, compared to applying water-soluble fertilizers only, which is coincident with the results reported by Cortina et al. [32], who noted that root growth in fertilized plants commonly increases when the phosphorus and nitrogen availability also increase, improving the ability of the plants to absorb water. Several authors have examined the favorable effects that increasing the dose of nutrients may have on growth and other desirable plant characteristics, both in the nursery and in the field [34,62–64]. Controlled-release fertilizers minimize the loss of nutrients by leaching, due to their potential to supply nutrients in synchrony with the needs of the plant [65] and in this work the plant responses, expressed in morphological and physiological quality indices, confirmed these effects (Table 3 and Figure 2).

## 5. Conclusions

The use of a larger dose of controlled-release fertilizer in the nursery, regardless of the substrates used, promoted greater plant growth responses. In general, at the high fertilizer rate, substrate composition had little significant effect of seedling morphology. At the lower fertilizer rate, a higher proportion of sawdust and lower proportion of peat moss tended to reduce seedling morphological attributes. Economic tradeoffs comparing higher use of readily available in-country resources (fertilizer, pine bark, sawdust) to the cost of imported resources (peat moss) were not part of this study, but should be considered by nursery managers when seeking alternative growing substrates.

**Author Contributions:** R.E.M.-A. was involved in the establishment of the experiment, collection, capture and statistical analysis of the data and the writing of the manuscript; J.Á.P.-R. participated in the design and establishment of the experiment and the reviewing of the manuscript; A.A. was involved in the experimental design and the reviewing of the manuscript; J.C.H.-D. was a counsellor on the development and results of the work, who reviewed and translated the manuscript; C.W. was involved in a review of the manuscript; J.A.C.-S. participated in a review of the manuscript; J.G.M. reviewed and translated the manuscript. All authors have read and agreed to the published version of the manuscript.

**Acknowledgments:** The authors wish to express their gratitude to Roberto Trujillo and Roberto Trujillo Ayala, who are managers of the "General Francisco Villa" Forest Nursery, for the facilities provided in order to carry out these experiments. Also, appreciation goes to Rodrigo Veliz López and José Vidales Trujillo, for the advice in the development of the experiment. Finally, thanks also go to CONACyT, for the doctoral scholarship granted to the first author.

**Conflicts of Interest:** The authors declare no conflict of interest.

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
