# Peer review of "Alternative Substrates and Fertilization Doses in the Production of Pinus cembroides Zucc. in Nursery"

_forests, doi:10.3390/f11010071_

Round 1

Reviewer 1 Report

I have no complaints about the revised manuscript.

Author Response

Response to the Reviewer 1.

Comments were attended and even in other sections of the text the wording and grammar were improved.

The manuscript was checked by a native English speaking colleague.  

Reviewer 2 Report

Comments on Forests 64287-V1

General Comments

This is a well-designed study of substrate composition crossed with fertilizer levels. The statistical analysis is reasonable, but I do think the authors should consider transforming the data to be able to use a parametric comparison and means separation test. The argument that sawdust results in seedlings with poorer root systems is not supported well enough by the data, and the experimental design is not intended to test this hypothesis. The inclusion of a tradeoff analysis would be important since this work is intended to support commercial nursery production of tree seedlings. The authors refer to it but do not include any data or analysis.

There are some small writing errors, especially the use of commas. I highlighted errors in my copy of the manuscript (attached) but did not make suggestions for changes. Good proofreading should be able to eliminate these errors.

Abstract

The authors use “F2” and “F3” to refer to fertilizer treatments, but the rest of the paper uses “F1” and “F2”.
The authors also refer to cost of treatments vs seedling performance, but there is no data to quantify this tradeoff.

Introduction

Line 58:                 Please include here the citation and reference to the original paper that proposed and tested what became known as the Dickson’s Quality Index.

Lines 84-86:          There is no need to try to list all the different ways fertilizer can be delivered to seedlings in the nursery. There are certainly more than 2 ways to do it, though.

Lines 76-83:          This is a reasonable introduction and justification for use pine sawdust for growing media, but there is no mention of composted bark in the introduction.

Materials and Methods

Lines 108-110:      If available, it would help to have a figure showing temperature and light conditions in the nursery during the experiment.

Lines 112-118:      If the seed treatment and germination procedures are based on standard and reported practices, it would help to have a citation and reference to that.

Lines 124-125:      Here and elsewhere, it would help to include the “%” symbol to make clear the number refers to the percent of a particular nutrient element the fertilizer.

Lines 134-140:      There is not enough information here to understand how much fertilizer was applied to the plants. It appears they fertigated the plants, but there is no mention of a schedule of fertigation or the total amount of water+fertilizer applied. Also, I assume “ME” refers to micronutrient elements, but the elements and concentrations should also be specified.

Lines 155-157:      Include citations and references to the elemental analysis methods.

Lines 165-171:      What was the Normality test used on the data?
Based on the distribution of the morphological data, there may be a justifiable transformation to achieve Normality and allow parametric comparisons so you don’t have to use such a low Type I error rate.

Discussion

Lines 246-249:      This statement needs to be reworded. It is unclear what “satisfactory field performance” refers to. The term “must” suggests the diameter limit is absolute or represents a threshold of performance, which is not described or justified.
Also, stating that a treatment “failed to achieve” the diameter limit is somewhat unclear. Are the authors referring to mean diameter or the maximum diameter of any seedling in this treatment? It would help to include “M1-M3” here so the correspondence with results in the table are clear.

Lines 244-288:      It’s not clear what the authors mean by “robustness of the plants” or “mitigate damage from drought or heat”. Including citations is not a sufficient explanation.
It is a reasonable hypothesis that pine sawdust is immobilizing N in the growing media, but there are no clear differences in N concentrations and biomass among the WF substrate treatments, so the analysis of results don’t support this or explain the difference in diameter.
The DQI results illustrate the mixed nature of the results.
The experimental design is not intended to isolate a single media component in explaining seedling performance, so the various lines of argument in this section are not sufficiently supported by the data.

Lines 301-311:      I’m not sure what the point of this section is, except that more fertilizer is better within the range evaluated here. There is no ability to evaluate the specific advantages of controlled release fertilizer vs liquid fertigation in this study. Leaching and nutrient availability were not measured, and there was no assessment of timing of release relative to plant demand. Thus, most of the implied advantages here were not specifically evaluated.

Conclusion

As with the abstract, the authors refer to tradeoff of performance with cost, but there is not data or analysis to support it.

Tables and Figures

Table 3:                 Please indicate what the numbers refer to in this table. They appear to be mean ± standard error, but this needs to be specified.

Figure 1:                The graphs are a bit small for the font size and resolution to show up well.

Author Response

Response letter reviewer 2

Ref: Forests-642987

Regarding the manuscript "Alternative substrates and fertilization doses in the production of Pinus cembroides Zucc. In the nursery" Ref: Forests-642987-MDPI, in this letter we describe the changes applied in the manuscript, where we attended the questions and incorporated the suggestions of the reviewers.

 Comments and suggestions of Reviewer 2

General comments

There are some small writing errors, especially the use of commas. I highlighted errors in my copy of the manuscript (attached) but did not make suggestions for changes. Good proofreading should be able to eliminate these errors.

Response: Comments were attended and even in other sections of the text the wording and grammar were improved.

The changes suggested are marked in red and those made to improve the writing are marked in turquoise.

The manuscript was checked by a native English speaking colleague. 

Abstract

Q1) The authors use “F2” and “F3” to refer to fertilizer treatments, but the rest of the paper uses “F1” and “F2”.

Response: F2 and F3 were corrected and replaced by F1 and F2 throughout the manuscript (Line 26,33,34).

Q2) The authors also refer to cost of treatments vs seedling performance, but there is no data to quantify this tradeoff.

Response: That part of the text was deleted, since the cost analysis was not included.

Introduction

Q3) Line 58: Please include here the citation and reference to the original paper that proposed and tested what became known as the Dickson’s Quality Index.

Response: The citation of the original document (line 58) was included.

Q4) Lines 84-86: There is no need to try to list all the different ways fertilizer can be delivered to seedlings in the nursery. There are certainly more than 2 ways to do it, though.

Response: The numbering was eliminated and these fertilizers are now mentioned as fertilization options (line 84-86).

Q5) Lines 76-83: This is a reasonable introduction and justification for use pine sawdust for growing media, but there is no mention of composted bark in the introduction.

Response: Now, the same paragraph indicates the importance of using composted bark as a substrate for plant production (Lines 76-78).

Materials and methods

Q6) Lines 108-110: If available, it would help to have a figure showing temperature and light conditions in the nursery during the experiment.

Response: A figure was added that includes temperature and relative humidity during plant growth (lines 112).

Q7) Lines 112-118: If the seed treatment and germination procedures are based on standard and reported practices, it would help to have a citation and reference to that.

Response: A citation was added which indicates the standardized procedure on seeds treatments, previous to and during their germination (line 120).

Q8) Lines 124-125: Here and elsewhere, it would help to include the “%” symbol to make clear the number refers to the percent of a particular nutrient element the fertilizer.

Response: The "%" symbol was included to indicate the percentage of each component of the fertilizers used (line 127 and 128).

Q9) Lines 134-140: There is not enough information here to understand how much fertilizer was applied to the plants. It appears they fertigated the plants, but there is no mention of a schedule of fertigation or the total amount of water+fertilizer applied. Also, I assume “ME” refers to micronutrient elements, but the elements and concentrations should also be specified.

Response: Information on how much fertilizer the plants received at each irrigation was included, as well as the amount of water applied per plant. In addition, the micronutrients contained in the fertilizers (ME) (lines 137-145) were specified.

Q10) Lines 155-157: Include citations and references to the elemental analysis methods.

Response: Citations describing the methods to determine nutrient contents (lines 161 and 163) were added.

Q11) Lines 165-171: What was the Normality test used on the data? Based on the distribution of the morphological data, there may be a justifiable transformation to achieve Normality and allow parametric comparisons so you don’t have to use such a low Type I error rate.

Response: Since the data did not meet the normality assumptions, according with the Shapiro–Wilk and Kolmogorov Smirnov tests, we tried to transform the variables with the square root function and using the arccosine function, but the results did not show normality. Therefore, the Kruskall Wallis non-parametric statistical test was used, with the PROC NPAR1WAY procedure.

Discussion

Q12) On line 918 in the revised text: Remove word ‘good’, because it has value loading.

Response: The word "good" was deleted (line 232).

Q13) Lines 246-249: This statement needs to be reworded. It is unclear what “satisfactory field performance” refers to. The term “must” suggests the diameter limit is absolute or represents a threshold of performance, which is not described or justified. Also, stating that a treatment “failed to achieve” the diameter limit is somewhat unclear. Are the authors referring to mean diameter or the maximum diameter of any seedling in this treatment? It would help to include “M1-M3” here so the correspondence with results in the table are clear.

Response: The paragraph was rewritten based on the reviewer's recommendations (lines 250-262).

Q14) Lines 244-288: It’s not clear what the authors mean by “robustness of the plants” or “mitigate damage from drought or heat”. Including citations is not a sufficient explanation.

Response: The paragraph was redrafted to give greater clarity by eliminating the term “plant robustness”, since it refers to the height (cm)/diameter (mm), which is a variable not estimated in this test (lines 250-262).

Q15) It is a reasonable hypothesis that pine sawdust is immobilizing N in the growing media, but there are no clear differences in N concentrations and biomass among the WF substrate treatments, so the analysis of results don’t support this or explain the difference in diameter.

Response: The paragraph was rewritten (lines 250-262).

Q16) The DQI results illustrate the mixed nature of the results. The experimental design is not intended to isolate a single media component in explaining seedling performance, so the various lines of argument in this section are not sufficiently supported by the data.

Response: Part of the paragraph was rewritten, where it is mentioned that the higher values of DQI coincide with the best results of the other variables, characterized by having the incorporation of both fertilizers (lines 272-277).

Q17) Lines 301-311: I’m not sure what the point of this section is, except that more fertilizer is better within the range evaluated here. There is no ability to evaluate the specific advantages of controlled release fertilizer vs liquid fertigation in this study. Leaching and nutrient availability were not measured, and there was no assessment of timing of release relative to plant demand. Thus, most of the implied advantages here were not specifically evaluated.

Response: The paragraph was rewritten. It is mentioned that the combined fertilization of slow-release fertilizers plus water-soluble fertilization favored the growth of plants in relation to when water-soluble fertilizer only was applied (lines 304-313).

Conclusion

Q18) As with the abstract, the authors refer to tradeoff of performance with cost, but there is not data or analysis to support it.

Response: The conclusions were rewritten to clarify the work’s reach (lines 315-322.

Tables and Figures

Q19) Table 3: Please indicate what the numbers refer to in this table. They appear to be mean ± standard error, but this needs to be specified.

Response: It was added the notation (mean ± standard error) in the Table’s heading (line 207).

Q20) Figure 1: The graphs are a bit small for the font size and resolution to show up well.

Response: The font size in Figure 2 was increased (lines 223).

Round 2

Reviewer 2 Report

The revisions are appropriate and address my major concerns. It is significantly improved.